# Gender Differences in the Extended Theory of Planned Behaviour on Smoking Cessation Intention in Young Soldiers

**DOI:** 10.3390/ijerph18157834

**Published:** 2021-07-23

**Authors:** Yi-Chun Liu, Li-Chen Yen, Fang-Yih Liaw, Ming-Han Lin, Shih-Hung Chiang, Fu-Gong Lin, Ching-Huang Lai, Senyeong Kao, Yu-Tien Chang, Chia-Chao Wu, Yu-Lung Chiu

**Affiliations:** 1Graduate Institute of Medical Sciences, National Defense Medical Center, Taipei City 114, Taiwan; liuyich@mail.ndmctsgh.edu.tw; 2Department of Microbiology and Immunology, National Defense Medical Center, Taipei City 114, Taiwan; yenlichen1030@gmail.com; 3Department of Family and Community Medicine, Tri-Service General Hospital, Taipei City 114, Taiwan; qqmug2000@hotmail.com; 4School of Public Health, National Defense Medical Center, Taipei City 114, Taiwan; tonylin258@gmail.com (M.-H.L.); han840402@yahoo.com.tw (S.-H.C.); fugong@mail.ndmctsgh.edu.tw (F.-G.L.); clai4330@gmail.com (C.-H.L.); kao@mail.ndmctsgh.edu.tw (S.K.); greengarden720925@gmail.com (Y.-T.C.); 5Graduate Institute of Life Sciences, National Defense Medical Center, Taipei City 114, Taiwan; 6Division of Nephrology, Department of Internal Medicine, Tri-Service General Hospital, National Defense Medical Center, Taipei City 114, Taiwan; wucc@ndmctsgh.edu.tw

**Keywords:** theory of planned behaviour (TPB), intention, smoking cessation, gender difference

## Abstract

Background: The theory of planned behaviour (TPB) explanation of smoking cessation intentions consists of gender differences. The purpose of this study is to adopt the extended TPB to discuss factors influencing the smoking cessation intentions of young adult volunteer soldiers and to further compare the respective factors for both genders. Methods: This is a cross-sectional study. Data were collected from 139 and 165 male and female volunteer soldiers who smoked, respectively. Research participants completed a self-administered questionnaire that comprised items pertaining to the participants’ demographic characteristics, smoking behaviours, smoking cessation experiences, social environments, and TPB variables. Results: Subjective norms (friends) are a positive key factor for young adult male (β = 0.033, *p* = 0.012) and female (β = 0.076, *p* < 0.001) volunteer soldiers’ smoking cessation intentions, and perceived behavioural control is a key factor for male young (β = 0.226, *p* = 0.040) adult volunteer soldiers’ smoking cessation intention. The extended TPB accounted for 27.9% and 53.2% of the variance in the intention to quit smoking in the male and female volunteer soldiers, respectively. Conclusions: We suggest that smoking cessation strategies can reinforce gender-specific intervention strategies to assist young adult volunteer soldiers in smoking cessation.

## 1. Introduction

The World Health Organization (WHO) declared smoking to be the top cause of preventable deaths in recent years. Annually, over seven million deaths are caused by smoking-related diseases worldwide. This number exceeds that of deaths caused by AIDS, malaria, and tuberculosis combined [1]. In addition, the United States Department of Health and Human Services (HHS) indicated that chronic exposure to the addictive substances in tobacco, chiefly, nicotine, results in cancer, respiratory diseases, and cardiovascular diseases as well as influencing the reproductive, nervous, and immune systems [2].

In Taiwan, volunteer soldiers are aged between 18 and 32, but they are a relatively disadvantaged group because of their lower education, lower household income, and higher percentage of indigenous people than the general public of the same age [3,4,5,6,7]. As Taiwan’s social environment changes, military personnel are retiring frequently, there are a declining birth rate and aging population, and recruited men are becoming gradually insufficient. Taiwan revolutionised its military service system from the transformation of the conscription system to the enlistment system from 2000, and the period of compulsory service was shortened from 2 years in 1990 to 4 months in 2013 [8]. From 2018, there has only been an enlistment system in Taiwan to recruit high-quality and long-term outstanding manpower to serve in the military [9].

Smoking lowers physical and professional performance for military personnel [10,11]. The smoking rates of male and female soldiers in Taiwan are 36.2% and 11.9%, respectively [12]. These numbers are higher than the smoking rates for Taiwanese men (20.2%) and women (3.7%) in the same age group [13]. Volunteer soldiers are allowed to smoke during the basic military training in Taiwan. However, there are many restrictions. For example, volunteer soldiers can smoke in the smoking-designated areas during limited times. In addition, reducing the number of designated smoking areas and lengthening the distance between the designated smoking areas increase inconvenience for smokers. However, despite the inconvenience created by their environments, during service, only 1.4–1.8% of soldiers successfully quitted smoking upon completion of recruit training [14], representing a lower smoking cessation efficiency than that of a 1 h US army health education course (14%) [15]. Research also indicated that only 20% of Taiwanese soldiers intended to quit smoking [16], a lower number than that for enlisted personnel at military bases in Spain [17]. Therefore, if we can increase the intention of 80% of those who do not have the intention to quit, and the success rate of those who have the intention to quit is 33% [18], the overall smoking rate may be reduced by 26.4%. As most soldiers are young adults, they are within the age group most likely to develop smoking behaviours [19]. Contrariwise, adults who quit smoking between 25 and 34 years old gained 10 years of life in comparison to those who continued to smoke [20]. Factors influencing young adults’ smoking cessation intentions may be used to increase their intentions to quit smoking, preventing smoking behaviours from becoming a lifelong habit and reducing smoking-related mortality risks.

As smoking rates in women are considerably lower than in men from Asian and Pacific Island countries, the problem of women smoking is often overlooked. However, because women’s smoking behaviours have evolved into a symbol of liberation and freedom, women’s smoking rates are on the rise [21]. Although women demonstrated stronger smoking cessation intentions than men, the smoking cessation success rate for women was lower than that for men. Multileveled influences may come from biological, psychological, and social factors, which change in different times and locations [22,23]. This indicates that comparisons between the smoking cessation intentions of different genders are beneficial to establishing tobacco control policies.

The theory of planned behaviour (TPB) [24] was developed from the theory of reasoned action (TRA) proposed by Fishbein and Ajzen (1977) [25]. Aside from TRA variables, namely, attitude (ATT) and subjective norms (SNs), the TPB also includes the perceived behavioural control (PBC) variable. In this context, ATT refers to individuals’ attitude towards behaviours and can be positive or negative; SNs describe individuals’ subjective perceptions of whether people closest to them approve of their behaviour; and PBC refers to the individuals’ perceptions of whether behaviours are achievable through self-regulation. Numerous studies have adopted the TPB to predict smoking cessation intention. Norwegian studies indicated that the smoking cessation intentions of teenagers were influenced by all three TPB variables. However, the smoking cessation intentions of adults between 35 and 55 years old were influenced only by ATT and PBC, with SNs exerting no influence [26,27]. Korean studies have revealed that the smoking cessation intentions of middle-aged adults were related only to ATT and SNs established by family members [28,29]. Taiwanese studies have indicated that the smoking cessation intentions of middle-aged adults were influenced by all three TPB variables [27,30]. The aforementioned studies have indicated that TPB variables exhibited different smoking cessation intentions predicting functions according to different research subjects. However, few studies adopted the TPB to examine the smoking cessation intentions of young adults. In addition, Dohnke et al. (2011) suggested that the TPB explanation of smoking cessation intentions consists of gender differences [31]. Therefore, this study further analysed the gender differences in adopting the TPB to explain the smoking cessation intentions of young adult volunteer soldiers.

Ajzen (1991) proposed the extended TPB theory, which includes additional variables to enhance the TPB model’s explanatory power for predicted behaviours [24]. To increase the TPB model’s explanatory power, this study inputted the following variables into the extended TPB model: smoking history, number of cigarettes consumed daily, smoke addiction level, number of days on which the participant smoked in the preceding month, the age at which the participant began smoking, whether the participant’s parents smoked, whether the participant’s friends smoked, whether the participant had already attempted to quit smoking before, and whether people who interacted with the participant supported the attempts to quit smoking (these variables were correlated with the participant’s individual smoking cessation intention) [18,27,28,29,31,32]. The purpose of this study was to apply the extended TPB to discuss the factors influencing the smoking cessation intentions of young adult volunteer soldiers and to further understand differences in the factors between genders.

## 2. Materials and Methods

### 2.1. Study Design and Participants

This is a cross-sectional study. The researchers conducted purposive sampling to select young adult volunteer soldiers from a recruit training unit between 1 June 2016 and 31 December 2017. The study accepted participants who had smoked in the preceding 30 days and accumulated at least 100 smoked cigarettes since starting smoking [33]. G-Power 3.1 software was employed to estimate the sample size required. This study adopted the multiple regression analysis method and set the following values: α = 0.05, power = 0.8, effect size = 0.15, and number of predictors = 15. The required sample size was 139 or more. The research protocol was approved by the Institutional Review Board of the Tri-Service General Hospital, National Defense Medical Center (1-105-05-034).

### 2.2. Measurements

After the researchers explained the research purpose and obtained the participants’ consent, the researchers collected information by using self-administered questionnaires. The questionnaire content validity was reviewed by health care and health care education experts. The questionnaire content included demographic characteristics, smoking behaviours and smoking cessation experiences, social environments, and TPB variables [27,28,29,30,34,35].

The demographic characteristics encompassed the participants’ ages, education levels (e.g., junior high school or lower, senior high school or vocational high school, and junior college or college and above), and marital statuses (married or unmarried).

Questions on smoking behaviours and smoking cessation experiences included smoking history, daily cigarette consumption, number of days on which the participant smoked in the preceding month, nicotine dependence (low, medium, or high), and whether the participant attempted to quit smoking within the previous year (yes or no). The nicotine dependence level was evaluated using the Chinese version of the Fagerström Test for Nicotine Dependence, which consists of six questions. The test scores (0–10) were classified into three levels, 0–3, 4–6, and 7–10, indicating whether the participant’s nicotine dependence was low, medium, or high, respectively [36].

The social environments questions included whether the participant was living with family members who smoked, whether the participant’s friends smoked, and whether the participant’s friends supported smoking cessation. Participants could respond with “no”, or “yes, less than half” or “yes, more than half”. If less than half of the participant’s friends or family members smoked, the participant answered “yes, less than half”; if more than half of the participant’s friends or family members smoked, the participant answered “yes, more than half”.

The TPB variables comprised ATT, SNs, PBC, and intention. The questionnaire referenced other studies to design the TPB variables questions [28,30,35]. The questionnaire consisted of four ATT items, each of which asked participants, “Smoking to you is ______.” Participant’s responses were recorded according to four 7-point adjective pairs, namely, good–bad, pleasant–unpleasant, beneficial–harmful, and interesting–boring. After calculating the average participant scores of the four items, Cronbach’s α was revealed to be 0.88. The SNs items consisted of two subscales that addressed family members and friends’ normative beliefs (NBs) and motivations to comply (MC). The participants’ responses were scored from 1 to 7, which represented strongly disagree to strongly agree, respectively. Participants’ SNs scores were calculated by multiplying the NBs scores with the MC scores. Cronbach’s α of the two NBs items, namely, “Do most [over half] of your close family members hope that you quit smoking?”, and “Do most [over half] of your close friends hope that you quit smoking?”, was 0.92. In comparison, the two MC items, namely, “Will you fulfil your family members’ expectations by quitting smoking?”, and “Will you fulfil your friends’ expectations by quitting smoking?”, demonstrated a Cronbach’s α value of 0.95. The PBC items established participants’ ability to resist smoking in the following eight conditions: when feeling nervous or anxious, when in a bad mood, after dining, when bored, when happy, when waking up, when encountering high-pressure events, and when facing encouragement to smoke by friends or peers. Each item was scored on a 7-point scale, (1 = strongly disagree; 7 = strongly agree), with the average score serving as the participants’ PBC score. Cronbach’s α was 0.99. Finally, the intention variable served as the main outcome variable and was evaluated using two items, namely, “Do you want to quit smoking within the next 6 months?”, and “Do you want to quit smoking within the next month?” The items were scored on a 7-point scale (1 = strongly disagree; 7 = strongly agree). The average score of the two items served as the total intention score. Cronbach’s α for the intention variable was 0.99.

### 2.3. Data Analysis

This study employed IBM SPSS Statistics V.22.0 (IBM, Armonk, NY, USA) to conduct statistical analysis. Continuous variables were displayed using means and standard deviations, whereas category variables were presented using frequencies and percentages. In addition, this study adopted independent *t* tests, one-way analysis of variance, and the Pearson correlation test to analyse the demographic characteristics, smoking behaviours and smoking cessation experiences, social environments, and the relationship between TPB variables and the outcome variable. Variables with a two-sided *p*-value of <0.05 in univariable analysis were considered significant factors influencing the outcome variable. Finally, the researchers adopted multiple linear regression analysis to analyse the factors influencing participants’ smoking cessation intentions. If the variance inflation factor was greater than or equal to 10, the variable was removed from the multiple linear regression analysis to avoid multicollinearity problems between the independent variables. A *p*-value < 0.05 (two-sided test) was considered statistically significant.

## 3. Results

### 3.1. Characteristics of Study Population

The participants consisted of 165 and 139 female and male volunteer soldiers, respectively. The average age of female participants was greater than that of male participants; the percentage of female participants with education up to the college level or higher was greater than the corresponding percentage for males; and the proportion of women who were married was lower than the corresponding proportion for men. Furthermore, female volunteer soldiers’ average smoking histories were significantly longer than those of males (5.02 vs. 4.09, *p* = 0.006); the average number of cigarettes consumed by female volunteer soldiers daily was less than for that for males (6.12 vs. 10.12, *p* < 0.001); and the average number of days on which female volunteer soldiers smoked in the preceding month (16.42) was significantly lower than that of males (24.27) (*p* < 0.001). Furthermore, 49.3% of female volunteer soldiers exhibited low-level, 38.8% medium-level, and 11.9% high-level nicotine dependency, proportions which were significantly greater than the corresponding proportions for males (41.8%, 32.8%, 25.4%) (*p* = 0.021). Finally, more female volunteer soldiers experienced smoking cessation support from people around them than males (76.7% vs. 65.0%, *p* = 0.011) (Table 1).

### 3.2. Difference between Males and Females in TBP Status

Mean scores for intention, SNs, and PBC were under the scale midpoint, but ATT was high. All the TPB factors of the female volunteer soldiers were greater than those for the males. However, only the average PBC score of female volunteer soldiers was significantly greater than that of males (Table 2).

### 3.3. Factors Associated with Smoking Cessation Intention

#### 3.3.1. Univariable Analysis

Table 3 shows that for the volunteer soldiers, the ATT, family SNs, friend SNs, and PBC scores were significantly and positively correlated with smoking cessation intention, but the number of days smoked in the preceding month was significantly and negatively correlated with smoking cessation intention.

By contrast, the female volunteer soldiers’ smoking histories and number of cigarettes consumed daily were significantly and negatively correlated with their smoking cessation intentions. Furthermore, for male volunteer soldiers, the smoking cessation intentions were significantly higher for those who never attempted to quit smoking within the previous year than for those who attempted to quit smoking. In addition, living with family members who smoked was related to intentions.

#### 3.3.2. Multivariable Analysis

Table 4 displays the multiple linear regression analysis results for the smoking cessation intention of young adult male and female volunteer soldiers. After controlling for relevant influential factors, the main influential factor of the smoking cessation intentions of female young adult volunteer soldiers was revealed to be friend SNs (β = 0.076, *p* < 0.001), and the comprehensive model explanatory power (adjusted R^2^) was 53.2%. Subsequently, after controlling for the relevant influential factors, the main factors influencing the smoking cessation intentions of male young adult volunteer soldiers were friend SNs (β = 0.033, *p* = 0.012) and PBC (β = 0.226, *p* = 0.040). The comprehensive model explanatory power (adjusted R^2^) was calculated to be 27.9%.

## 4. Discussion

We employed the extended TPB to discuss the gender-specific influential factors of smoking cessation intentions for young adult volunteer soldiers. In the future, the study results can serve as a reference to establish gender-specific smoking cessation interventions for young adults. This study revealed that the average smoking cessation intention scores of female and male volunteer soldiers were not significantly different, and both were lower than the scale’s median score. This result is consistent with other studies [26,28,37].

Among all the TPB variables for both genders, only the average PBC score was significantly higher for female volunteer soldiers than for males. Possible reasons for this result include the lower cigarette consumption and smoking dependency exhibited by female volunteer soldiers than by males, which would endow females with more self-control to resist smoking [38].

However, no gender difference existed regarding the SNs of family members and friends. This finding echoes the findings of other studies, according to which, although the acceptance of women smoking is lower in Asian countries than in other countries, young age groups have become more accepting towards women smoking [39]. This finding also resembles that of Dohnke et al. (2011) [31], indicating that female young adults did not notably perceive negative views from society.

Meta-analysis studies indicated that TPB variables exhibited an average explanatory power of 39% for behaviour intention [40]. However, the extended model constructed in this study demonstrated 27.9% and 53.2% explanatory power for the smoking cessation intentions of male and female volunteer soldiers, respectively, indicating that the extended model demonstrated greater suitability in applications to understand the smoking cessation intentions of female volunteer soldiers.

The SNs of friends were related to the smoking cessation intentions of both male and female young adult volunteer soldiers. This effect was revealed to be stronger for females than for males, supporting the findings of a study conducted on 16- to 19-year-old participants in Norway [26]. However, among Korean American middle-aged women and men, the SNs of friends had no relationship with the smoking cessation intentions [28,29]. This implies that female soldiers need more support from other people to quit smoking than males, and that the gender differences in SNs were more related to age than to culture.

In both genders, the SNs from family members did not influence the participants’ smoking cessation intentions. This result contradicts the findings of studies on Korean Americans [28,29]. However, the range of participants’ ages for the study on Korean Americans was between 42 and 47 years, older than the participants of this study. This indicates that, to young adults, the influence of friends is greater than that of family members. This finding is consistent with that of other studies that suggests the smoking cessation behaviours of young adults to be influenced by peers [41].

This study also reveals that PBC was exclusively related to male young adult volunteer soldiers’ smoking cessation intentions and served as the most influential factor among the TPB variables. However, PBC did not influence the smoking cessation intentions of female young adult volunteer soldiers. This finding resembles the results of Høie et al. (2012) [26]. Despite female volunteer soldiers exhibiting significantly higher PBC scores than males, PBC exerted a greater smoking cessation intention influence on male young adult volunteer soldiers. This suggests that male young adult volunteer soldiers with stronger self-control exhibited a stronger smoking cessation intention.

The one-way analysis of variance indicated that ATT is positively related to smoking cessation intention. In the TPB variables, ATT demonstrated a lower correlation coefficient with smoking cessation intention. However, after controlling the TPB variables, the relationship between ATT and smoking cessation intention disappeared. This result corresponds to that of other studies [42]. Possible reasons for this outcome include young adults’ awareness of the risks associated with smoking. Therefore, despite exhibiting positive attitudes to quitting smoking, the smoking cessation intentions of the young adults from this study were not increased.

In the smoking behaviour variables, experience in quitting smoking and the smoking behaviours of family members and friends were unrelated to smoking cessation intention. This suggests that smoking cessation intentions were more related to TPB variables. Alternatively, this may result from the shorter smoking histories of participants in this study, which caused the smoking behaviours of young adults to be less affected by those of family members and friends.

This study may serve as a reference for designing young adult, gender-specific smoking cessation education to increase their intention to quit smoking. However, the following limitations apply: (1) This was a cross-sectional study, and the proposed TPB variables did not exhibit causal effects on smoking cessation intention. Follow-up research is suggested to confirm the causal relationship. (2) As this study employed in-service volunteer soldiers as participants, the findings of this study may not necessarily apply to young adults from other fields. (3) The PBC items in the questionnaire asked questions regarding the participants’ ability to resist smoking in eight different conditions instead of directly asking the participants questions regarding their ability to quit smoking. However, studies have already verified that the ability to resist smoking is reliably able to predict smoking cessation intention [30]. Therefore, this does not affect the inference relationship between PBC and smoking cessation intention.

## 5. Conclusions

The study results indicate that the extended TPB can effectively predict the smoking cessation intention of young adult volunteer soldiers, and that SNs from friends are related to smoking cessation intention. However, PBC is exclusively related to the smoking cessation intention of male young adult volunteer soldiers. The study results imply that future designs of smoking cessation education must consider gender-specific viewpoints. Aside from encouraging friends and peers to quit smoking together, intervention methods can adopt health care education methods to teach males how to resist smoking, thereby effectively increasing the smoking cessation intention of males. In addition, future studies can further verify the relationship between TPB variables and smoking cessation behaviours.

## Figures and Tables

**Table 1 ijerph-18-07834-t001:** Demographic characteristics, smoking behaviours, smoking cessation experiences, and social environments of study population.

Variables	Female (n = 165)	Male (n = 139)	*p*-Value
	n (%)/Mean ± SD	n (%)/Mean ± SD	
Demographics			
Age	23.04 ± 3.76	20.59 ± 2.20	<0.001 *
Education level			0.002 *
Junior high school or lower	13 (8.0)	1 (0.7)	
Senior high school or vocational high school	129 (79.1)	127 (92.0)	
Junior college or college and above	21 (12.9)	10 (7.3)	
Marital statuses			0.039 *
Married	158 (95.8)	138 (100)	
Unmarried	7 (4.2)	0 (0)	
Smoking behaviours and smoking cessation experiences			
Smoking history	5.02 ± 3.34	4.09 ± 2.45	0.006 *
Daily cigarette consumption	6.12 ± 6.54	10.12 ± 7.73	<0.001 *
Number of days smoked in the preceding month	16.42 ± 10.81	24.27 ± 8.87	<0.001 *
Nicotine dependence			0.021 *
Low	66 (49.3)	51 (41.8)	
Medium	52 (38.8)	40 (32.8)	
High	16 (11.9)	31 (25.4)	
Attempted to quit smoking within the previous year			0.773
No	85 (52.8)	69 (51.1)	
Yes	76 (47.2)	66 (48.9)	
Social environments			
Living with family members who smoke			0.156
No	37 (22.7)	41 (29.5)	
Yes, less than half	93 (57.1)	80 (57.6)	
Yes, more than half	33 (20.2)	18 (12.9)	
Friends who smoke			0.976
No	4 (2.5)	4 (2.9)	
Yes, less than half	79 (49.7)	69 (50.0)	
Yes, more than half	76 (47.8)	65 (47.1)	
Smoking cessation supported by people			0.011 *
No	38 (23.3)	48 (35.0)	
Yes, less than half	81 (49.7)	69 (50.4)	
Yes, more than half	44 (27.0)	20 (14.6)	

* *p*-value < 0.05.

**Table 2 ijerph-18-07834-t002:** Mean, SD for TPB variables.

Variables	Female	Male	*p*-Value
	Mean ± SD	Mean ± SD	
Intention ^a^	3.18 ± 1.93	2.95 ± 1.88	0.298
ATT ^a^	4.42 ± 1.47	4.38 ± 1.81	0.823
SNs (family members) ^b^	24.48 ± 14.85	23.88 ± 14.02	0.721
SNs (friends) ^b^	18.39 ± 14.89	16.83 ± 13.52	0.348
PBC ^a^	2.93 ± 1.51	2.48 ± 1.41	0.009 *

TPB, theory of planned behaviour; ATT = attitude, SNs = subjective norms, PBC = perceived behavioural control. ^a^ Ranges from 1 to 7. ^b^ Ranges from 1 to 49. * *p*-value < 0.05.

**Table 3 ijerph-18-07834-t003:** Univariable analysis of TPB, demographics, smoking, and social environments on smoking cessation intention by gender.

Variables	Smoking Cessation Intention					
	Female			Male		
	Mean ± SD	t/F/r	*p*-Value	Mean ± SD	t/F/r	*p*-Value
TPB						
ATT ^a^		0.393	<0.001 *		0.281	0.001 *
SNs (family members) ^b^		0.523	<0.001 *		0.443	<0.001 *
SNs (friends) ^b^		0.700	<0.001 *		0.446	<0.001 *
PBC ^a^		0.341	<0.001 *		0.343	<0.001 *
Demographics						
Age		−0.041	0.603		−0.162	0.059
Education level		0.511	0.601		0.981	0.377
Junior high school or lower	2.69 ± 2.29			1.00 ± N/A		
Senior high school or vocational high school	3.24 ± 1.93			2.92 ± 1.85		
Junior college or college and above	3.05 ± 1.83			3.50 ± 2.30		
Marital statuses		0.954	0.341			
Married	3.21 ± 1.91			2.94 ± 1.88		
Unmarried	2.50 ± 2.47			-		
Smoking behaviours and smoking cessation experiences						
Smoking history		−0.271	<0.001 *		−0.163	0.060
Daily cigarette consumption		−0.173	0.027 *		−0.160	0.062
Number of days smoked in the preceding month		−0.322	<0.001 *		−0.211	0.013 *
Nicotine dependence		0.877	0.418		1.426	0.245
Low	3.08 ± 1.75			3.17 ± 1.88		
Medium	2.99 ± 1.90			2.90 ± 1.79		
High	2.41 ± 1.90			2.43 ± 1.99		
Attempted to quit smoking within the previous year		1.183	0.239		2.213	0.029 *
No	3.37 ± 1.87			3.34 ± 1.92		
Yes	3.00 ± 1.97			2.63 ± 1.79		
Social environments						
Living with family members who smoke		2.420	0.092		7.098	0.035 *
No	3.21 ± 1.98			3.65 ± 2.04		
Yes, less than half	3.36 ± 1.89			2.87 ± 1.78		
Yes, more than half	2.52 ± 1.88			1.75 ± 1.17		
Friends who smoke		2.787	0.068		2.333	0.101
No	3.13 ± 2.46			4.00 ± 2.04		
Yes, less than half	3.52 ± 1.93			3.17 ± 1.96		
Yes, more than half	2.80 ± 1.85			2.59 ± 1.68		
Smoking cessation supported by people		1.980	0.141		0.122	0.886
No	2.88 ± 2.00			2.90 ± 1.71		
Yes, less than half	3.05 ± 1.80			3.04 ± 2.00		
Yes, more than half	3.65 ± 2.01			2.85 ± 1.89		

TPB, theory of planned behaviour; ATT = attitude, SNs = subjective norms, PBC = perceived behavioural control. ^a^ Ranges from 1 to 7. ^b^ Ranges from 1 to 49. * *p*-value < 0.05.

**Table 4 ijerph-18-07834-t004:** Multivariable linear regression of smoking cessation intention.

Variables	Smoking Cessation Intention			
	Female		Male	
	β (95%CI)	*p*-Value	β (95%CI)	*p*-Value
TPB				
ATT ^a^	0.081 (−0.091, 0.253)	0.353	0.001 (−0.174, 0.176)	0.991
SNs (family members) ^b^	0.006 (−0.016, 0.027)	0.593	0.025 (−0.001, 0.051)	0.057
SNs (friends) ^b^	0.076 (0.054, 0.099)	<0.001 *	0.033 (0.007, 0.058)	0.012 *
PBC ^a^	0.153 (−0.005, 0.310)	0.058	0.226 (0.010, 0.441)	0.040 *
Smoking behaviours and smoking cessation experiences				
Smoking history	0.006 (−0.068, 0.079)	0.881		
Daily cigarette consumption	−0.016 (−0.052, 0.020)	0.378		
Number of days smoked in the preceding month	−0.021 (−0.043, 0.001)	0.063	−0.018 (−0.051, 0.015)	0.284
Attempted to quit smoking within the previous year				
No			ref.	
Yes			0.187 (−0.387, 0.761)	0.521
Social environment				
Living with family members who smoke				
No			ref.	
Yes, less than half			−0.317 (−0.962, 0.327)	0.331
Yes, more than half			−0.731 (−1.703, 0.241)	0.139
	*p* < 0.001		*p* < 0.001	
	R^2^ = 0.554, adjusted R^2^ = 0.532		R^2^ = 0.325, adjusted R^2^ = 0.279	

TPB, theory of planned behaviour; ATT = attitude, SNs = subjective norms, PBC = perceived behavioural control. ^a^ Ranges from 1 to 7. ^b^ Ranges from 1 to 49. * *p*-value < 0.05.

## Data Availability

The data presented in this study are available on request from the corresponding author. The data are not publicly available due to privacy.

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
