# Peer review of "Gender Differences in the Extended Theory of Planned Behaviour on Smoking Cessation Intention in Young Soldiers"

_ijerph, 2021, doi:10.3390/ijerph18157834_

Round 1
Reviewer 1 Report
This paper represents a significant contribution to the literature and may be especially important for understanding factors associated with smoking cessation among military personnel, a group for whom smoking is particularly common. As indicated by the authors, the gender differences identified in relation to intention to quit smoking may be useful for informing the design of smoking cessation programming.
The paper would be improved significantly if the authors were more explicit about the nature of the study population and how it is similar/different from the general population of young adults in Taiwan and globally. The study participants are volunteer military personnel and as such the findings may not be generalizable to a more general young adult population. The authors note this in the limitations, yet the conclusions are stated in such a way that the results may be informative for a more broad young adult population. It is my understanding that Taiwan transitioned to a volunteer military relatively recently. It would be helpful to include some details related to this transition in the introduction along with some description of how this population is similar/different from the general young adult population so that readers can make more confident assessments about generalizability of findings presented. Furthermore, gender differences in Taiwan and other Asian nations may not be representative of those in other countries such as the US. The authors should be more explicit about the populations referenced in other studies (e.g., line 280 - reference #18 points to a study of Korean participants) and also address the fact that findings may not generalize to other nations with different gender norms, broadly and specific to smoking behaviors.
Additionally, the results section of the paper is somewhat confusing as currently presented. The tables (4 & 5) should be referenced in the text and the paragraphs included in this section (beginning on lines 232 and 239 respectively) should also include specific reference to the analytic model being described. At present, it is challenging for the reader to understand the difference between the two models.
A number of minor edits are needed to address issues related to language. Examples include:
Line 219 - smoked should be smoking and additional info. is needed following the word negatively
Line225 - who should be those
Line 231 - multivariable is misspelled
Attention to the issues described will substantially improve the paper and ensure that it will be suitable for publication.
Author Response
Point 1: The paper would be improved significantly if the authors were more explicit about the nature of the study population and how it is similar/different from the general population of young adults in Taiwan and globally. The study participants are volunteer military personnel and as such the findings may not be generalizable to a more general young adult population. The authors note this in the limitations, yet the conclusions are stated in such a way that the results may be informative for a more broad young adult population. It is my understanding that Taiwan transitioned to a volunteer military relatively recently. It would be helpful to include some details related to this transition in the introduction along with some description of how this population is similar/different from the general young adult population so that readers can make more confident assessments about generalizability of findings presented.
Response 1: Thank you for your comments. We have included details about the nature of the volunteer soldiers and evolution of the Military Service System in Taiwan in the Introduction as follows: “In Taiwan, volunteer soldiers aged between 18 and 32, but they are a relatively disadvantaged group because of their lower education, lower household income and higher percentage of indigenous people than the general public of the same age [1,2,3,4,5]. As Taiwan’s social environment changes, military personnel retired frequently, the declining birth rate and aging population and the gradually insufficient recruited men, Taiwan had the revolution of military service system from the transformation of conscription system to enlistment system from 2000, and the period of compulsory service was shortened from 2 years on 1990 to 4 months on 2013 [6]. From 2018, there was only enlistment system in Taiwan to recruit high-quality and long-term outstanding manpower to serve in the military [7].” (2nd paragraph, page 2)
References
- Cheng, K.C. (2021). Association between e-cigarettes use, smoking and sleep quality among volunteer soldiers at the basic military training center. Unpublished master dissertation, National Defense Medical Center, Taipei, Taiwan.
- Chang, H.T. (2019). Exploring Quits & Win smoking cessation application’s continuance intention to use of volunteer military on Technology Acceptance Model. Unpublished master dissertation, National Defense Medical Center, Taipei, Taiwan.
- Department of Statistics, Ministry of the interior, R.O.C. (Taiwan). 2020 Statistical Bulletin on week 12. Available online: https://www.moi.gov.tw/cp.aspx?n=10492 (accessed on 6 July 2021).
- Directorate-General of Budget, Accounting and Statistics, Executive Yuan, R.O.C. (Taiwan). Report on the Survey of Family Income and Expenditure, 2019. [CrossRef]
- Department of Statistics, Ministry of the interior, R.O.C. (Taiwan). 2021 Statistical Bulletin on week 18. Available online: https://www.moi.gov.tw/cl.aspx?n=13331 (accessed on 6 July 2021).
- Hong, J.C.; Shih, I-H. History, Change, and Outlook of R.O.C’s Military Service System. Archives Quarterly 2013, 12, 30-45. [CrossRef]
- Ministry of National Defense, R.O.C. (Taiwan). Available online: https://www.mnd.gov.tw/ (accessed on 6 July 2021).
Point 2: Furthermore, gender differences in Taiwan and other Asian nations may not be representative of those in other countries such as the US. The authors should be more explicit about the populations referenced in other studies (e.g., line 280 - reference #18 points to a study of Korean participants) and also address the fact that findings may not generalize to other nations with different gender norms, broadly and specific to smoking behaviors.
Response 2: Thank you for your comments. The research we cited was from Norway. Hence, the gender differences in Taiwan were similar to Norway. However, among Korean American women and men, the SNs of friends had no relationship with the smoking cessation intentions. Therefore, we infer that gender differences in SN are more related to age than to culture. As a result, we have revised the paragraph in the Discussion as follows: “The SNs of friends were related to the smoking cessation intentions of both male and female young adult volunteer soldiers. This effect was revealed to be stronger for female than for male, supporting the findings of a study conducted on 16- to 19-year-old participants in Norway [26]. However, among Korean American middle-aged women and men, the SNs of friends had no relationship with the smoking cessation intentions [28,29]. This imply that female soldiers need more support from other people to quit smoking than male and the gender differences in SN were more related to age than to culture.” (5th paragraph, page 8)
Point 3: Additionally, the results section of the paper is somewhat confusing as currently presented. The tables (4 & 5) should be referenced in the text and the paragraphs included in this section (beginning on lines 232 and 239 respectively) should also include specific reference to the analytic model being described. At present, it is challenging for the reader to understand the difference between the two models.
Response 3: Thank you for your comments. We have referenced tables 4 in the text and added specific reference to the analytic model in the paragraphs included in this section. According to reviewer 2’s strong recommendation and our main analysis in table 4, we have modified to present only one final model (table 4) and deleted table 5 in the original submission. We modified the paragraph in the Result as follows: “Table 4 displays the multiple linear regression analysis results for the smoking cessation intention of young adult male and female volunteer soldiers.” (1st paragraph, page 7)
Point 4: A number of minor edits are needed to address issues related to language. Examples include:
Line 219 - smoked should be smoking and additional info. is needed following the word negatively
Response 4: Thank you for your kind reminder. We have modified the sentence in the 3.3.1. Univariable Analysis as follows: “Table 3 displays that for the volunteer soldiers, the ATT, family SN, friend SN, and PBC scores were significantly and positively correlated to smoking cessation intention, but number of days smoked in the preceding month was significantly and negatively correlated to smoking cessation intention.” (2nd paragraph, page 6)
Point 5: Line225 - who should be those
Response 5: Thank you for your kind reminder. We have modified the sentence in the 3.3.1. Univariable Analysis as follows: “Furthermore, for male volunteer soldiers, the smoking cessation intentions were significantly higher for who never attempted to quit smoking within the previous year than for those who attempted to quit smoking. In addition, living with family members smoked was related to the intentions.” (3rd paragraph, page 6)
Point 6: Line 231 - multivariable is misspelled
Response 6: Thank you for your kind reminder. We have modified the title as “Multivariable Analysis” (1st title, page 7)
ijerph-1262319 manuscript with track changes, please see the attachment.

Reviewer 2 Report
This is a cross-sectional study exploring gender differences in the extended theory of planned behavior (eTBP) on smoking cessation intention in young soldiers in Taiwan. This is an interesting study, it is well-written and relatively easy to follow. However, it has both major and minor issues that should be addressed.
MAJOR ISSUES:
As I find this study both interesting and relevant, I really miss a paragraph mentioning the clinical or societal relevance of this study, alongside an estimate of the impact of the study, e.g. how and to what extent could this study impact the smoking prevalence among potentially how many young soldiers in Taiwan?
Methods (data analysis):
As you are running a lot of statistical tests (28 in table 2 alone), you should explain how you dealt with the risk of multiplicity.
*****
In the measurement section you mention intention as you main outcome. In the analysis section you use the term "outcome variables" (line 180). Please specify all outcomes.
*****
Please elaborate on this section e.g.
- Did you use one-sided or two-sided tests and what was you level of significance?
- How did you plan to conduct your regression analysis (e.g. stepwise backward/enter all factors together ect.)
- Please specify which "relevant influential factors" you adjusted for and how you reached that decision.
Results:
Please add references to the tables in the text.
*****
Below I have illustrated a general issue in the results section using this example:
Page 4, line 196-198: Furthermore, 88.1% of female volunteer soldiers exhibited low- and medium-level nicotine dependency, which was significantly greater than the corresponding proportion for male (74.6%) (P = .021).
As I understand you performed a chi2-test on a 2x3 table. In this text you collapse your categories from 3 to 2 categories (low/mediun AND high) and then use the p-value from the 2x3 test to determine the significance! If you want to state results on collapsed categories, you should run the test on a 2x2 table instead - in this case it would still yield a significant p-value of 0.025. Alternatively just mentions the numbers of the collapsed categories and don't use the test result.
*****
Page 6, line 225 ff: ... of who living with family members did not smoke were significantly greater than those living with family members smoked.
(Please correct the language,) and you might be right, but couldn't the difference be due to differences between the "Yes less than half" and "Yes more than half"?
*****
3.3.2 Miltivariable Analysis (Multivariable Analysis)
If I understand correctly you are actually performing two multivariable analysis using different approaches? One where you enter all variables together, and one where you use a stepwise (backward/forward?) method? As there is no reference to the tables in the text it is unclear when you refer to what table.
If this is the case, it is unclear to my why both methods are used instead of just one (which you should have decided on on forehand). As I see it you don't really use the second analysis (table 5), therefore I assume your main analysis is the one presented in table 4! I strongly recommend that you only present one final model.
If I have misunderstood the setup, than please argue why you have chosen to use two different approaches.
Discussion:
Page 9, line 310: This study may serve as reference for gender-specific smoking cessation interventions aimed at young adults.
I am curious as to how you reach this conclusion. As I see it the models were not that different across gender. Also your main outcome was smoking cessation intention and you conclude on smoking cessation intervention, could you please elaborate on differences/similarities/association between the two, that justifies this conclusion.
Conclusion:
Please see my comment above. As mentioned I don't see the models as that different, and again I'm curious about how you go from measuring intention to concluding on intervention.
MINOR ISSUES:
In general:
Even though I find the manuscript quite easy to read, I also stumbled upon sentences which I found difficult to understand. A few examples are: in the description of a variable: "number of days on which smoking was engaged in during the preceding month"; or from the discussion: "It may be more smoking cessation support from people for female than male soldiers".
Please read the manuscript carefully to clean out this sentences.
Introduction:
Page 2, line 95: Therefore, researchers further analysed...
It is unclear to me whether you are referring to yourselfs and this study, or somebody else work. Please clarify.
Results:
Page 6, line 219: ... were significantly negatively.
Please clarify! (I assume it's the correlation that is negative and significant, but is is unclear).
Author Response
Point 1: MAJOR ISSUES:
As I find this study both interesting and relevant, I really miss a paragraph mentioning the clinical or societal relevance of this study, alongside an estimate of the impact of the study, e.g. how and to what extent could this study impact the smoking prevalence among potentially how many young soldiers in Taiwan?
Response 1: Thank you for your comments. A study in Taiwan have indicated that only 20% of soldiers intended to quit smoking [1]. And a study performed by Moan and Rise (2005) found that 33% of those who were strongly motivated to quit smoking subsequently succeeded in quitting [2]. Therefore, if we can increase the intention of 80% of those who do not have the intention to quit, the overall smoking rate may be reduced by 26.4%. We have supplemented the contents in the 1. Introduction as follows: “Research also indicated that only 20% of Taiwanese soldiers intended to quit smoking [16], a lower number than for enlisted personnel at military bases in Spain [17]. Therefore, if we can increase the intention of 80% of those who do not have the intention to quit, and the success rate of those who have the intention to quit is 33% [18], the overall smoking rate may be reduced by 26.4%.” (3rd paragraph, page 2)
References
- Tai, Z.; Tao, S.P.; Hung, Y.J. Cigarette use, smoking cessation, and quit intentions among active-duty military personnel in Taiwan. Mil Psychol 2012, 24, 236–250. [CrossRef]
- Moan, I.S.; Rise, J. Quitting Smoking: Applying an Extended Version of the Theory of Planned Behavior to Predict Intention and Behavior. J Appl Biobehav Res 2005, 10, 39–68. [CrossRef]
Point 2: Methods (data analysis):
As you are running a lot of statistical tests (28 in table 2 alone), you should explain how you dealt with the risk of multiplicity.
Response 2: Thank you for kindly reminder. In Table 3, we tested the relationship between independent variables and one outcome (intention to smoking cessation). We have modified the presentation of the table 3 to avoid misunderstandings. (table 3, page 6)
Table 3. Univariable analysis of TPB, demographics, smoking and social environments on intention to smoking cessation by gender.
|
Variables |
Smoking cessation intention |
|
||||
|
Female |
Male |
|
||||
|
Mean SD |
t/F/r |
p value |
Mean SD |
t/F/r |
p value |
|
|
TPB |
|
|
|
|
|
|
|
ATT a |
|
0.393 |
<0.001* |
|
0.281 |
0.001* |
|
SN (family members) b |
|
0.523 |
<0.001* |
|
0.443 |
<0.001* |
|
SN (friends) b |
|
0.700 |
<0.001* |
|
0.446 |
<0.001* |
|
PBC a |
|
0.341 |
<0.001* |
|
0.343 |
<0.001* |
|
Demographics |
|
|
|
|
|
|
|
Age |
|
-0.041 |
0.603 |
|
-0.162 |
0.059 |
|
Education level |
|
0.511 |
0.601 |
|
0.981 |
0.377 |
|
Junior high school or lower |
2.69 2.29 |
|
|
1.00 N/A |
|
|
|
Senior high school or vocational high school |
3.24 1.93 |
|
|
2.92 1.85 |
|
|
|
Junior college or college and above |
3.05 1.83 |
|
|
3.50 2.30 |
|
|
|
Marital statuses |
|
0.954 |
0.341 |
|
|
|
|
Married |
3.21 1.91 |
|
|
2.94 1.88 |
|
|
|
Unmarried |
2.50 2.47 |
|
|
- |
|
|
|
Smoking behaviours and smoking cessation experiences |
|
|
|
|
|
|
|
Smoking history |
|
-0.271 |
<0.001* |
|
-0.163 |
0.060 |
|
Daily cigarette consumption |
|
-0.173 |
0.027* |
|
-0.160 |
0.062 |
|
Number of days smoked in the preceding month |
|
-0.322 |
<0.001* |
|
-0.211 |
0.013* |
|
Nicotine dependence |
|
0.877 |
0.418 |
|
1.426 |
0.245 |
|
Low |
3.08 1.75 |
|
|
3.17 1.88 |
|
|
|
Medium |
2.99 1.90 |
|
|
2.90 1.79 |
|
|
|
High |
2.41 1.90 |
|
|
2.43 1.99 |
|
|
|
Attempted to quit smoking within the previous year |
|
1.183 |
0.239 |
|
2.213 |
0.029* |
|
No |
3.37 1.87 |
|
|
3.34 1.92 |
|
|
|
Yes |
3.00 1.97 |
|
|
2.63 1.79 |
|
|
|
Social environments |
|
|
|
|
|
|
|
Living with family members smoked |
|
2.420 |
0.092 |
|
7.098 |
0.035* |
|
No |
3.21 1.98 |
|
|
3.65 2.04 |
|
|
|
Yes, less than half |
3.36 1.89 |
|
|
2.87 1.78 |
|
|
|
Yes, more than half |
2.52 1.88 |
|
|
1.75 1.17 |
|
|
|
Friends smoked |
|
2.787 |
0.068 |
|
2.333 |
0.101 |
|
No |
3.13 2.46 |
|
|
4.00 2.04 |
|
|
|
Yes, less than half |
3.52 1.93 |
|
|
3.17 1.96 |
|
|
|
Yes, more than half |
2.80 1.85 |
|
|
2.59 1.68 |
|
|
|
Smoking cessation supported from people |
|
1.980 |
0.141 |
|
0.122 |
0.886 |
|
No |
2..88 2.00 |
|
|
2.90 1.71 |
|
|
|
Yes, less than half |
3.05 1.80 |
|
|
3.04 2.00 |
|
|
|
Yes, more than half |
3.65 2.01 |
|
|
2.85 1.89 |
|
|
TPB, Theory of Planned Behaviour; ATT = attitude, SN = subjective norms, PBC = perceived behavioural control. a Ranges from 1 to 7. b Ranges from 1 to 49. * p value <0.05.
Point 3: In the measurement section you mention intention as you main outcome. In the analysis section you use the term "outcome variables" (line 180). Please specify all outcomes.
Response 3: Thank you for your kind reminder. There was only one dependent variable in our study. We have modified the sentence in the 2.3. Data Analysis as follows: “In addition, this study adopted independent t tests, one-way analysis of variance, and the Pearson correlation test to analyse the demographic characteristics, smoking behaviours and smoking cessation experiences, social environments, and the relationship between TPB variables and outcome variable.” (4th paragraph, page 4)
Point 4: Please elaborate on this section e.g.
Did you use one-sided or two-sided tests and what was you level of significance? How did you plan to conduct your regression analysis (e.g. stepwise backward/enter all factors together ect.) Please specify which "relevant influential factors" you adjusted for and how you reached that decision.
Response 4: Thank you for your comments. This study adopted the two-sided tests method and the level of significance was set at p < 0.05. According to your variable suggestion, we decided that the regression analysis was analysed using enter method. And variables with a two-sided p-value of < 0.05 in univariable analysis were considered significant factors adjusted in the regression analysis. We have supplemented the contents in the 2.3. Data Analysis as follows: “Variables with a two-sided p-value of < 0.05 in univariable analysis were considered significant factors influencing outcome variable. A p-value < 0.05 (two-sided test) was considered statistically significant.” (4th paragraph, page 4)
Point 5: Results:
Please add references to the tables in the text.
Response 5: Thank you for your kind reminder. We have added references to the table in the text. We placed the paragraph in the 3.3.2. Multivariable Analysis as follows: “Table 4 displays the multiple linear regression analysis results for the smoking cessation intention of young adult male and female volunteer soldiers.” (1st paragraph, page 7)
Point 6: Below I have illustrated a general issue in the results section using this example:
Page 4, line 196-198: Furthermore, 88.1% of female volunteer soldiers exhibited low- and medium-level nicotine dependency, which was significantly greater than the corresponding proportion for male (74.6%) (P = .021).
As I understand you performed a chi2-test on a 2x3 table. In this text you collapse your categories from 3 to 2 categories (low/mediun AND high) and then use the p-value from the 2x3 test to determine the significance! If you want to state results on collapsed categories, you should run the test on a 2x2 table instead - in this case it would still yield a significant p-value of 0.025. Alternatively just mentions the numbers of the collapsed categories and don't use the test result.
Response 6: Thank you for your comment. We have modified the sentence in the 3.1. Characteristics of Study Population as follows: “Furthermore, 49.3% of female volunteer soldiers exhibited low-level, 38.8% medium-level and 11.9% high-level nicotine dependency, which was significantly greater than the corresponding proportion for male (41.8%, 32.8%, 25.4%) (P = .021).” (1st paragraph, page 5)
Point 7: Page 6, line 225 ff: ... of who living with family members did not smoke were significantly greater than those living with family members smoked.
(Please correct the language,) and you might be right, but couldn't the difference be due to differences between the "Yes less than half" and "Yes more than half"?
Response 7: Thank you for your comment. We have modified the sentence in the 3.3.1. Univariable Analysis as follows: “Furthermore, for male volunteer soldiers, the smoking cessation intentions were significantly higher for who never attempted to quit smoking within the previous year than for those who attempted to quit smoking. In addition, living with family members smoke was related to the intentions.” (3rd paragraph, page 6)
Point 8: 3.3.2 Miltivariable Analysis (Multivariable Analysis)
If I understand correctly you are actually performing two multivariable analysis using different approaches? One where you enter all variables together, and one where you use a stepwise (backward/forward?) method? As there is no reference to the tables in the text it is unclear when you refer to what table.
If this is the case, it is unclear to my why both methods are used instead of just one (which you should have decided on forehand). As I see it you don't really use the second analysis (table 5), therefore I assume your main analysis is the one presented in table 4! I strongly recommend that you only present one final model.
If I have misunderstood the setup, than please argue why you have chosen to use two different approaches.
Response 8: Thank you for your comments. According to your strong recommendation, we have modified to present only one final model (table 4), deleted table 5 in the original submission and added reference to the table 4 in the text.
Point 9:
Discussion:
Page 9, line 310: This study may serve as reference for gender-specific smoking cessation interventions aimed at young adults.
I am curious as to how you reach this conclusion. As I see it the models were not that different across gender. Also your main outcome was smoking cessation intention and you conclude on smoking cessation intervention, could you please elaborate on differences/similarities/association between the two, that justifies this conclusion.
Conclusion:
Please see my comment above. As mentioned I don't see the models as that different, and again I'm curious about how you go from measuring intention to concluding on intervention.
Response 9: Thank you for your comments. We found that the different factor influencing smoking cessation intentions among female and male volunteer soldiers was the perceived behavioural control (PBC). The PBC only significantly correlated to smoking cessation intentions among male volunteer soldiers. We have modified the sentence in the 5. Conclusion as follows: “The study results indicate that extended TPB can effectively predict smoking cessation intention of young adult volunteer soldiers, that SNs from friends are related to smoking cessation intention. However, PBC is exclusively related to the smoking cessation intention of young male adult volunteer soldiers.” (6th paragraph, page 9)
Thank you for your reminder. We have revised the sentence in the 4. Discussion as follows: “This study may serve as reference for designing young adults’ gender-specific smoking cessation education to increase their intention to quit smoking.” (5th paragraph, page 9) And the sentence in the 5. Conclusion as follows: “The study results imply that future designs of smoking cessation education must consider gender-specific viewpoints.” (6th paragraph, page 9)
Point 10: MINOR ISSUES:
In general:
Even though I find the manuscript quite easy to read, I also stumbled upon sentences which I found difficult to understand. A few examples are: in the description of a variable: "number of days on which smoking was engaged in during the preceding month"; or from the discussion: "It may be more smoking cessation support from people for female than male soldiers".
Please read the manuscript carefully to clean out this sentences.
Response 10: Thank you for your comments. We have modified the description of a variable in the 1. Introduction as follows: "number of days smoked in the preceding month" (2nd paragraph, page 3) and revised the sentence in the 4. Discussion as follows: “This imply that female soldiers need more support from other people to quit smoking than male and the gender differences in SN were more related to age than to culture.” (5th paragraph, page 8)
Point 11: Introduction:
Page 2, line 95: Therefore, researchers further analysed...
It is unclear to me whether you are referring to yourselfs and this study, or somebody else work. Please clarify.
Response 11: Thank you for your comments. We have modified the sentence in the 1. Introduction as follows: “Therefore, this study further analysed the gender differences in adopting TPB to explain the smoking cessation intentions of young adult volunteer soldiers.” (1st paragraph, page 3)
Point 12: Results:
Page 6, line 219: ... were significantly negatively.
Please clarify! (I assume it's the correlation that is negative and significant, but it is unclear)..
Response 12: Thank you for your comments. We have modified the sentence in the 3.3.1. Univariable Analysis as follows: “Table 3 displays that for the volunteer soldiers, the ATT, family SN, friend SN, and PBC scores were significantly and positively correlated to smoking cessation intention, but the number of days smoked in the preceding month was significantly and negatively correlated to smoking cessation intention.” (2nd paragraph, page 6)
